# One-shot learning by inverting a compositional causal process

**Brenden M. Lake**
Dept. of Brain and Cognitive Sciences
MIT
brenden@mit.edu

**Ruslan Salakhutdinov**
Dept. of Statistics and Computer Science
University of Toronto
rsalakhu@cs.toronto.edu

**Joshua B. Tenenbaum**
Dept. of Brain and Cognitive Sciences
MIT
jbt@mit.edu

## Abstract

People can learn a new visual class from just one example, yet machine learning algorithms typically require hundreds or thousands of examples to tackle the same problems. Here we present a Hierarchical Bayesian model based on compositionality and causality that can learn a wide range of natural (although simple) visual concepts, generalizing in human-like ways from just one image. We evaluated performance on a challenging one-shot classification task, where our model achieved a human-level error rate while substantially outperforming two deep learning models. We also tested the model on another conceptual task, generating new examples, by using a "visual Turing test" to show that our model produces human-like performance.

## 1 Introduction

People can acquire a new concept from only the barest of experience – just one or a handful of examples in a high-dimensional space of raw perceptual input. Although machine learning has tackled some of the same classification and recognition problems that people solve so effortlessly, the standard algorithms require hundreds or thousands of examples to reach good performance. While the standard MNIST benchmark dataset for digit recognition has 6000 training examples per class [19], people can classify new images of a foreign handwritten character from just one example (Figure 1b) [23, 16, 17]. Similarly, while classifiers are generally trained on hundreds of images per class, using benchmark datasets such as ImageNet [4] and CIFAR-10/100 [14], people can learn a

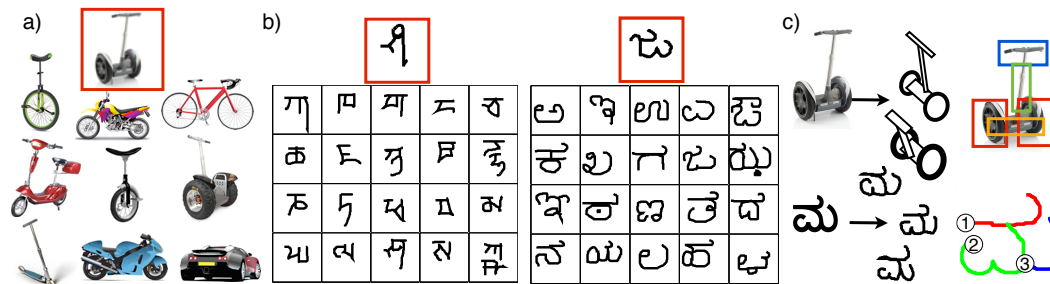

Figure 1: Can you learn a new concept from just one example? (a & b) Where are the other examples of the concept shown in red? Answers for b) are row 4 column 3 (left) and row 2 column 4 (right). c) The learned concepts also support many other abilities such as generating examples and parsing.

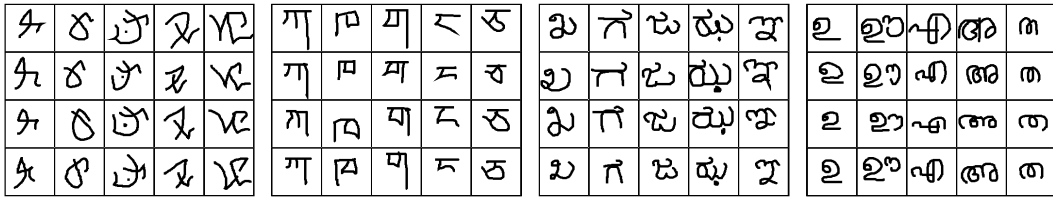

Figure 2: Four alphabets from Omniglot, each with five characters drawn by four different people.

new visual object from just one example (e.g., a "Segway" in Figure 1a). These new larger datasets have developed along with larger and "deeper" model architectures, and while performance has steadily (and even spectacularly [15]) improved in this big data setting, it is unknown how this progress translates to the "one-shot" setting that is a hallmark of human learning [3, 22, 28].

Additionally, while classification has received most of the attention in machine learning, people can generalize in a variety of other ways after learning a new concept. Equipped with the concept "Segway" or a new handwritten character (Figure 1c), people can produce new examples, parse an object into its critical parts, and fill in a missing part of an image. While this flexibility highlights the richness of people's concepts, suggesting they are much more than discriminative features or rules, there are reasons to suspect that such sophisticated concepts would be difficult if not impossible to learn from very sparse data. Theoretical analyses of learning express a tradeoff between the complexity of the representation (or the size of its hypothesis space) and the number of examples needed to reach some measure of "good generalization" (e.g., the bias/variance dilemma [8]). Given that people seem to succeed at both sides of the tradeoff, a central challenge is to explain this remarkable ability: What types of representations can be learned from just one or a few examples, and how can these representations support such flexible generalizations?

To address these questions, our work here offers two contributions as initial steps. First, we introduce a new set of one-shot learning problems for which humans and machines can be compared side-by-side, and second, we introduce a new algorithm that does substantially better on these tasks than current algorithms. We selected simple visual concepts from the domain of handwritten characters, which offers a large number of novel, high-dimensional, and cognitively natural stimuli (Figure 2). These characters are significantly more complex than the simple artificial stimuli most often modeled in psychological studies of concept learning (e.g., [6, 13]), yet they remain simple enough to hope that a computational model could see all the structure that people do, unlike domains such as natural scenes. We used a dataset we collected called "Omniglot" that was designed for studying learning from a few examples [17, 26]. While similar in spirit to MNIST, rather than having 10 characters with 6000 examples each, it has over 1600 character with 20 examples each – making it more like the "transpose" of MNIST. These characters were selected from 50 different alphabets on www.omniglot.com, which includes scripts from natural languages (e.g., Hebrew, Korean, Greek) and artificial scripts (e.g., Futurama and ULOG) invented for purposes like TV shows or video games. Since it was produced on Amazon's Mechanical Turk, each image is paired with a movie ([$x$,$y$,time] coordinates) showing how that drawing was produced.

In addition to introducing new one-shot learning challenge problems, this paper also introduces Hierarchical Bayesian Program Learning (HBPL), a model that exploits the principles of compositionality and causality to learn a wide range of simple visual concepts from just a single example. We compared the model with people and other competitive computational models for character recognition, including Deep Boltzmann Machines [25] and their Hierarchical Deep extension for learning with very few examples [26]. We find that HBPL classifies new examples with near human-level accuracy, substantially beating the competing models. We also tested the model on generating new exemplars, another natural form of generalization, using a "visual Turing test" to evaluate performance. In this test, both people and the model performed the same task side by side, and then other human participants judged which result was from a person and which was from a machine.

## 2   Hierarchical Bayesian Program Learning

We introduce a new computational approach called Hierarchical Bayesian Program Learning (HBPL) that utilizes the principles of compositionality and causality to build a probabilistic generative model of handwritten characters. It is compositional because characters are represented as stochastic motor programs where primitive structure is shared and re-used across characters at multiple levels, including strokes and sub-strokes. Given the raw pixels, the model searches for a

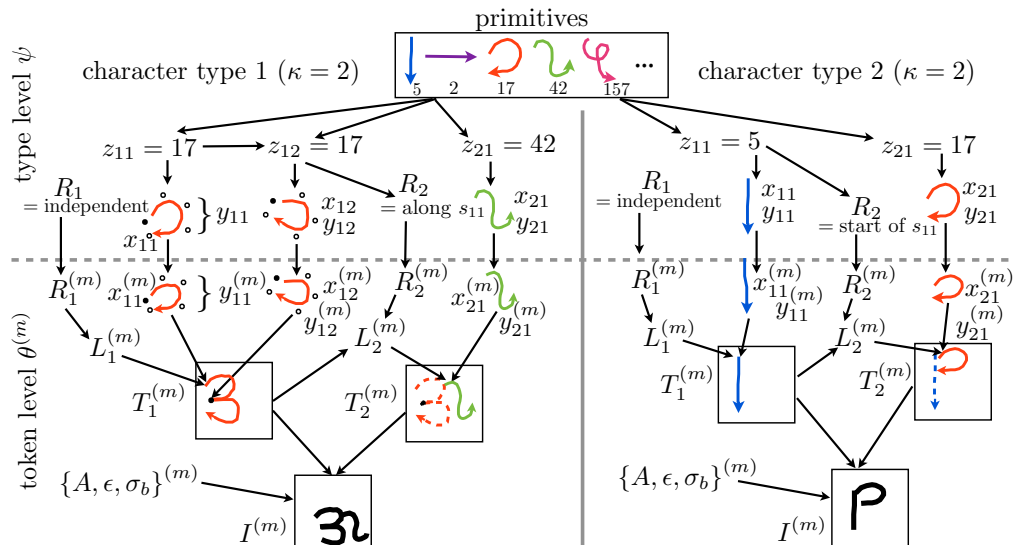

Figure 3: An illustration of the HBPL model generating two character types (left and right), where the dotted line separates the type-level from the token-level variables. Legend: number of strokes $\kappa$, relations $R$, primitive id $z$ (color-coded to highlight sharing), control points $x$ (open circles), scale $y$, start locations $L$, trajectories $T$, transformation $A$, noise $\epsilon$ and $\theta_b$, and image $I$.

"structural description" to explain the image by freely combining these elementary parts and their spatial relations. Unlike classic structural description models [27, 2], HBPL also reflects abstract causal structure about how characters are actually produced. This type of causal representation is psychologically plausible, and it has been previously theorized to explain both behavioral and neuro-imaging data regarding human character perception and learning (e.g., [7, 1, 21, 11, 12, 17]). As in most previous "analysis by synthesis" models of characters, strokes are not modeled at the level of muscle movements, so that they are abstract enough to be completed by a hand, a foot, or an airplane writing in the sky. But HBPL also learns a significantly more complex representation than earlier models, which used only one stroke (unless a second was added manually) [24, 10] or received on-line input data [9], sidestepping the challenging parsing problem needed to interpret complex characters.

The model distinguishes between character types (an 'A', 'B', etc.) and tokens (an 'A' drawn by a particular person), where types provide an abstract structural specification for generating different tokens. The joint distribution on types $\psi$, tokens $\theta^{(m)}$, and binary images $I^{(m)}$ is given as follows,

$$P(\psi, \theta^{(1)}, ..., \theta^{(M)}, I^{(1)}, ..., I^{(M)}) = P(\psi) \prod_{m=1}^{M} P(I^{(m)}|\theta^{(m)})P(\theta^{(m)}|\psi). \qquad (1)$$

Pseudocode to generate from this distribution is shown in the Supporting Information (Section SI-1).

## 2.1 Generating a character type

A character type $\psi = \{\kappa, S, R\}$ is defined by a set of $\kappa$ strokes $S = \{S_1, ..., S_\kappa\}$ and spatial relations $R = \{R_1, ..., R_\kappa\}$ between strokes. The joint distribution can be written as

$$P(\psi) = P(\kappa) \prod_{i=1}^{\kappa} P(S_i)P(R_i|S_1, ..., S_{i-1}). \qquad (2)$$

The number of strokes is sampled from a multinomial $P(\kappa)$ estimated from the empirical frequencies (Figure 4b), and the other conditional distributions are defined in the sections below. All hyperparameters, including the library of primitives (top of Figure 3), were learned from a large "background set" of character drawings as described in Sections 2.3 and SI-4.

**Strokes.** Each stroke is initiated by pressing the pen down and terminated by lifting the pen up. In between, a stroke is a motor routine composed of simple movements called substrokes $S_i = \{s_{i1}, ..., s_{in_i}\}$ (colored curves in Figure 3), where sub-strokes are separated by

brief pauses of the pen. Each sub-stroke $s_{ij}$ is modeled as a uniform cubic b-spline, which can be decomposed into three variables $s_{ij} = \{z_{ij}, x_{ij}, y_{ij}\}$ with joint distribution $P(S_i) = P(z_i) \prod_{j=1}^{n_i} P(x_{ij}|z_{ij})P(y_{ij}|z_{ij})$. The discrete class $z_{ij} \in \mathbb{N}$ is an index into the library of primitive motor elements (top of Figure 3), and its distribution $P(z_i) = P(z_{i1}) \prod_{j=2}^{n_i} P(z_{ij}|z_{i(j-1)})$ is a first-order Markov Process that adds sub-strokes at each step until a special "stop" state is sampled that ends the stroke. The five control points $x_{ij} \in \mathbb{R}^{10}$ (small open circles in Figure 3) are sampled from a Gaussian $P(x_{ij}|z_{ij}) = N(\mu_{z_{ij}}, \Sigma_{z_{ij}})$, but they live in an abstract space not yet embedded in the image frame. The type-level scale $y_{ij}$ of this space, relative to the image frame, is sampled from $P(y_{ij}|z_{ij}) = \text{Gamma}(\alpha_{z_{ij}}, \beta_{z_{ij}})$.

**Relations.** The spatial relation $R_i$ specifies how the beginning of stroke $S_i$ connects to the previous strokes $\{S_1, ..., S_{i-1}\}$. The distribution $P(R_i|S_1, ..., S_{i-1}) = P(R_i|z_1, ..., z_{i-1})$, since it only depends on the number of sub-strokes in each stroke. Relations can come in four types with probabilities $\theta_R$, and each type has different sub-variables and dimensionalities:

- *Independent* relations, $R_i = \{J_i, L_i\}$, where the position of stroke $i$ does not depend on previous strokes. The variable $J_i \in \mathbb{N}$ is drawn from $P(J_i)$, a multinomial over a 2D image grid that depends on index $i$ (Figure 4c). Since the position $L_i \in \mathbb{R}^2$ has to be real-valued, $P(L_i|J_i)$ is then sampled uniformly at random from within the image cell $J_i$.
- *Start* or *End* relations, $R_i = \{u_i\}$, where stroke $i$ starts at either the beginning or end of a previous stroke $u_i$, sampled uniformly at random from $u_i \in \{1, ..., i-1\}$.
- *Along* relations, $R_i = \{u_i, v_i, \tau_i\}$, where stroke $i$ begins along previous stroke $u_i \in \{1, ..., i-1\}$ at sub-stroke $v_i \in \{1, ..., n_{u_i}\}$ at type-level spline coordinate $\tau_i \in \mathbb{R}$, each sampled uniformly at random.

## 2.2 Generating a character token

The token-level variables, $\theta^{(m)} = \{L^{(m)}, x^{(m)}, y^{(m)}, R^{(m)}, A^{(m)}, \sigma_b^{(m)}, \epsilon^{(m)}\}$, are distributed as

$$P(\theta^{(m)}|\psi) = P(L^{(m)}|\theta_{\backslash L^{(m)}}^{(m)}, \psi) \prod_i P(R_i^{(m)}|R_i)P(y_i^{(m)}|y_i)P(x_i^{(m)}|x_i)P(A^{(m)}, \sigma_b^{(m)}, \epsilon^{(m)})$$

(3)

with details below. As before, Sections 2.3 and SI-4 describe how the hyperparameters were learned.

**Pen trajectories.** A stroke trajectory $T_i^{(m)}$ (Figure 3) is a sequence of points in the image plane that represents the path of the pen. Each trajectory $T_i^{(m)} = f(L_i^{(m)}, x_i^{(m)}, y_i^{(m)})$ is a deterministic function of a starting location $L_i^{(m)} \in \mathbb{R}^2$, token-level control points $x_i^{(m)} \in \mathbb{R}^{10}$, and token-level scale $y_i^{(m)} \in \mathbb{R}$. The control points and scale are noisy versions of their type-level counterparts, $P(x_{ij}^{(m)}|x_{ij}) = N(x_{ij}, \sigma_x^2 I)$ and $P(y_{ij}^{(m)}|y_{ij}) \propto N(y_{ij}, \sigma_y^2)$, where the scale is truncated below 0. To construct the trajectory $T_i^{(m)}$ (see illustration in Figure 3), the spline defined by the scaled control points $y_1^{(m)} x_1^{(m)} \in \mathbb{R}^{10}$ is evaluated to form a trajectory,[1] which is shifted in the image plane to begin at $L_i^{(m)}$. Next, the second spline $y_2^{(m)} x_2^{(m)}$ is evaluated and placed to begin at the end of the previous sub-stroke's trajectory, and so on until all sub-strokes are placed.

Token-level relations must be exactly equal to their type-level counterparts, $P(R_i^{(m)}|R_i) = \delta(R_i^{(m)} - R_i)$, except for the "along" relation which allows for token-level variability for the attachment along the spline using a truncated Gaussian $P(\tau_i^{(m)}|\tau_i) \propto N(\tau_i, \sigma_\tau^2)$. Given the pen trajectories of the previous strokes, the start position of $L_i^{(m)}$ is sampled from $P(L_i^{(m)}|R_i^{(m)}, T_1^{(m)}, ..., T_{i-1}^{(m)}) = N(g(R_i^{(m)}, T_1^{(m)}, ..., T_{i-1}^{(m)}), \Sigma_L)$, where $g(\cdot) = L_i$ when $R_i^{(m)}$ is *independent* (Section 2.1), $g(\cdot) = \text{end}(T_{u_i}^{(m)})$ or $g(\cdot) = \text{start}(T_{u_i}^{(m)})$ when $R_i^{(m)}$ is *start* or *end*, and $g(\cdot)$ is the proper spline evaluation when $R_i^{(m)}$ is *along*.

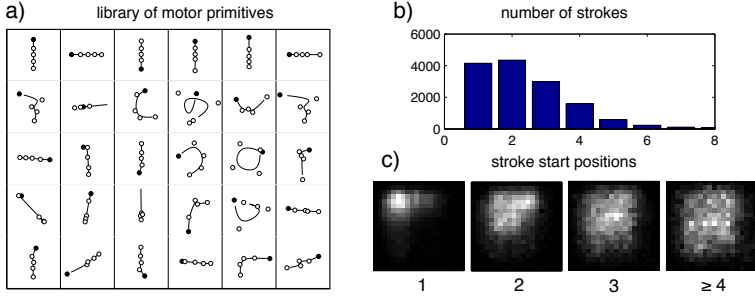

Figure 4: Learned hyperparameters. a) A subset of primitives, where the top row shows the most common ones. The first control point (circle) is a filled. b&c) Empirical distributions where the heatmap c) show how starting point differs by stroke number.

**Image.** An image transformation $A^{(m)} \in R^4$ is sampled from $P(A^{(m)}) = N([1,1,0,0], \Sigma_A)$, where the first two elements control a global re-scaling and the second two control a global translation of the center of mass of $T^{(m)}$. The transformed trajectories can then be rendered as a 105x105 grayscale image, using an ink model adapted from [10] (see Section SI-2). This grayscale image is then perturbed by two noise processes, which make the gradient more robust during optimization and encourage partial solutions during classification. These processes include convolution with a Gaussian filter with standard deviation $\sigma_b^{(m)}$ and pixel flipping with probability $\epsilon^{(m)}$, where the amount of noise $\sigma_b^{(m)}$ and $\epsilon^{(m)}$ are drawn uniformly on a pre-specified range (Section SI-2). The grayscale pixels then parameterize 105x105 independent Bernoulli distributions, completing the full model of binary images $P(I^{(m)}|\theta^{(m)}) = P(I^{(m)}|T^{(m)}, A^{(m)}, \sigma_b^{(m)}, \epsilon^{(m)})$.

### 2.3 Learning high-level knowledge of motor programs

The Omniglot dataset was randomly split into a 30 alphabet "background" set and a 20 alphabet "evaluation" set, constrained such that the background set included the six most common alphabets as determined by Google hits. Background images, paired with their motor data, were used to learn the hyperparameters of the HBPL model, including a set of 1000 primitive motor elements (Figure 4a) and position models for a drawing's first, second, and third stroke, etc. (Figure 4c). Wherever possible, cross-validation (within the background set) was used to decide issues of model complexity within the conditional probability distributions of HBPL. Details are provided in Section SI-4 for learning the models of primitives, positions, relations, token variability, and image transformations.

### 2.4 Inference

Posterior inference in this model is very challenging, since parsing an image $I^{(m)}$ requires exploring a large combinatorial space of different numbers and types of strokes, relations, and sub-strokes. We developed an algorithm for finding $K$ high-probability parses, $\psi^{[1]}, \theta^{(m)[1]}, ..., \psi^{[K]}, \theta^{(m)[K]}$, which are the most promising candidates proposed by a fast, bottom-up image analysis, shown in Figure 5a and detailed in Section SI-5. These parses approximate the posterior with a discrete distribution,

$$P(\psi, \theta^{(m)}|I^{(m)}) \approx \sum_{i=1}^{K} w_i \delta(\theta^{(m)} - \theta^{(m)[i]}) \delta(\psi - \psi^{[i]}), \qquad (4)$$

where each weight $w_i$ is proportional to parse score, marginalizing over shape variables $x$,

$$w_i \propto \tilde{w}_i = P(\psi_{\setminus x}^{[i]}, \theta^{(m)[i]}, I^{(m)}) \qquad (5)$$

and constrained such that $\sum_i w_i = 1$. Rather than using just a point estimate for each parse, the approximation can be improved by incorporating some of the local variance around the parse. Since the token-level variables $\theta^{(m)}$, which closely track the image, allow for little variability, and since it is inexpensive to draw conditional samples from the type-level $P(\psi|\theta^{(m)[i]}, I^{(m)}) = P(\psi|\theta^{(m)[i]})$ as it does not require evaluating the likelihood of the image, just the local variance around the type-level is estimated with the token-level fixed. Metropolis Hastings is run to produce N samples (Section SI-5.5) for each parse $\theta^{(m)[i]}$, denoted by $\psi^{[i1]}, ..., \psi^{[iN]}$, where the improved approximation is

$$P(\psi, \theta^{(m)}|I^{(m)}) \approx Q(\psi, \theta^{(m)}, I^{(m)}) = \sum_{i=1}^{K} w_i \delta(\theta^{(m)} - \theta^{(m)[i]}) \frac{1}{N} \sum_{j=1}^{N} \delta(\psi - \psi^{[ij]}). \qquad (6)$$

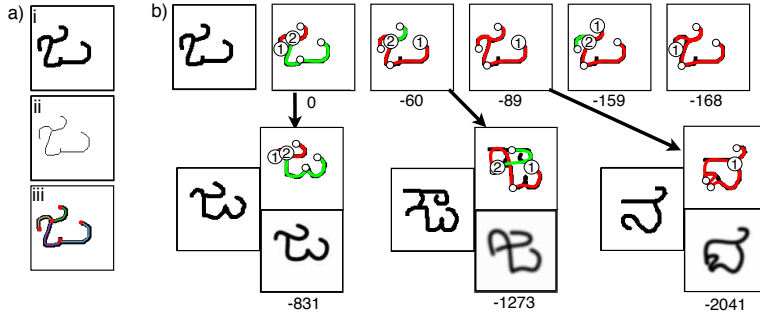

Figure 5: Parsing a raw image. a) The raw image (i) is processed by a thinning algorithm [18] (ii) and then analyzed as an undirected graph [20] (iii) where parses are guided random walks (Section SI-5). b) The five best parses found for that image (top row) are shown with their $\log w_j$ (Eq. 5), where numbers inside circles denote stroke order and starting position, and smaller open circles denote sub-stroke breaks. These five parses were re-fit to three different raw images of characters (left in image triplets), where the best parse (top right) and its associated image reconstruction (bottom right) are shown above its score (Eq. 9).

Given an approximate posterior for a particular image, the model can evaluate the posterior predictive score of a new image by re-fitting the token-level variables (bottom Figure 5b), as explained in Section 3.1 on inference for one-shot classification.

# 3 Results

## 3.1 One-shot classification

People, HBPL, and several alternative models were evaluated on a set of 10 challenging one-shot classification tasks. The tasks tested within-alphabet classification on 10 alphabets, with examples in Figure 2 and detailed in Section SI-6 . Each trial (of 400 total) consists of a single test image of a new character compared to 20 new characters from the same alphabet, given just one image each produced by a typical drawer of that alphabet. Figure 1b shows two example trials.

**People.** Forty participants in the USA were tested on one-shot classification using Mechanical Turk. On each trial, as in Figure 1b, participants were shown an image of a new character and asked to click on another image that shows the same character. To ensure classification was indeed "one shot," participants completed just one randomly selected trial from each of the 10 within-alphabet classification tasks, so that characters never repeated across trials. There was also an instructions quiz, two practice trials with the Latin and Greek alphabets, and feedback after every trial.

**Hierarchial Bayesian Program Learning**. For a test image $I^{(T)}$ and 20 training images $I^{(c)}$ for $c = 1, ..., 20$, we use a Bayesian classification rule for which we compute an approximate solution

$$\operatorname*{argmax}_{c} \ \log P(I^{(T)}|I^{(c)}). \tag{7}$$

Intuitively, the approximation uses the HBPL search algorithm to get $K = 5$ parses of $I^{(c)}$, runs $K$ MCMC chains to estimate the local type-level variability around each parse, and then runs $K$ gradient-based searches to re-optimizes the token-level variables $\theta^{(T)}$ (all are continuous) to fit the test image $I^{(T)}$. The approximation can be written as (see Section SI-7 for derivation)

$$\log P(I^{(T)}|I^{(c)}) \ \approx \ \log \int P(I^{(T)}|\theta^{(T)})P(\theta^{(T)}|\psi)Q(\theta^{(c)}, \psi, I^{(c)}) \, \mathrm{d}\psi \, \mathrm{d}\theta^{(c)} \, \mathrm{d}\theta^{(T)} \tag{8}$$

$$\approx \ \log \sum_{i=1}^{K} w_i \max_{\theta^{(T)}} P(I^{(T)}|\theta^{(T)}) \frac{1}{N} \sum_{j=1}^{N} P(\theta^{(T)}|\psi^{[ij]}), \tag{9}$$

where $Q(\cdot, \cdot, \cdot)$ and $w_i$ are from Eq. 6. Figure 5b shows examples of this classification score. While inference so far involves parses of $I^{(c)}$ refit to $I^{(T)}$, it also seems desirable to include parses of $I^{(T)}$ refit to $I^{(c)}$, namely $P(I^{(c)}|I^{(T)})$. We can re-write our classification rule (Eq. 7) to include just the reverse term (Eq. 10 center), and then to include both terms (Eq. 10 right), which is the rule we use,

$$\operatorname*{argmax}_{c} \ \log P(I^{(T)}|I^{(c)}) = \operatorname*{argmax}_{c} \ \log \frac{P(I^{(c)}|I^{(T)})}{P(I^{(c)})} = \operatorname*{argmax}_{c} \ \log \frac{P(I^{(c)}|I^{(T)})}{P(I^{(c)})} P(I^{(T)}|I^{(c)}), \tag{10}$$

where $P(I^{(c)}) \approx \sum_i \tilde{w}_i$ from Eq. 5. These three rules are equivalent if inference is exact, but due to our approximation, the two-way rule performs better as judged by pilot results.

**Affine model.** The full HBPL model is compared to a transformation-based approach that models the variance in image tokens as just global scales, translations, and blur, which relates to congealing models [23]. This HBPL model "without strokes" still benefits from good bottom-up image analysis (Figure 5) and a learned transformation model. The Affine model is identical to HBPL during search, but during classification, only the warp $A^{(m)}$, blur $\sigma_b^{(m)}$, and noise $\epsilon^{(m)}$ are re-optimized to a new image (change the argument of "max" in Eq. 9 from $\theta^{(T)}$ to $\{A^{(T)}, \sigma_b^{(T)}, \epsilon^{(T)}\}$).

**Deep Boltzmann Machines (DBMs).** A Deep Boltzmann Machine, with three hidden layers of 1000 hidden units each, was generatively pre-trained on an enhanced background set using the approximate learning algorithm from [25]. To evaluate classification performance, first the approximate posterior distribution over the DBMs top-level features was inferred for each image in the evaluation set, followed by performing 1-nearest neighbor in this feature space using cosine similarity. To speed up learning of the DBM and HD models, the original images were down-sampled, so that each image was represented by 28x28 pixels with greyscale values from [0,1]. To further reduce overfitting and learn more about the 2D image topology, which is built in to some deep models like convolution networks [19], the set of background characters was artificially enhanced by generating slight image translations (+/- 3 pixels), rotations (+/- 5 degrees), and scales (0.9 to 1.1).

**Hierarchical Deep Model (HD).** A more elaborate Hierarchical Deep model is derived by composing hierarchical nonparametric Bayesian models with Deep Boltzmann Machines [26]. The HD model learns a hierarchical Dirichlet process (HDP) prior over the activities of the top-level features in a Deep Boltzmann Machine, which allows one to represent both a layered hierarchy of increasingly abstract features and a tree-structured hierarchy of super-classes for sharing abstract knowledge among related classes. Given a new test image, the approximate posterior over class assignments can be quickly inferred, as detailed in [26].

**Simple Strokes (SS).** A much simpler variant of HBPL that infers rigid "stroke-like" parts [16].

**Nearest neighbor (NN).** Raw images are directly compared using cosine similarity and 1-NN.

**Results.** Performance is summarized in Table 1. As predicted, people were skilled one-shot learners, with an average error rate of 4.5%. HBPL achieved a similar error rate of 4.8%, which was significantly better than the alternatives. The Affine model achieved an error rate of 18.2% with the classification rule in Eq. 10 left, while performance was 31.8% error with Eq. 10 right. The deep learning models performed at 34.8% and 38% error, although performance was much lower without pre-training (68.3% and 72%). The Simple Strokes and Nearest Neighbor models had the highest error rates.

Table 1: One-shot classifiers

| Learner | Error rate |
|---|---|
| Humans | 4.5% |
| HBPL | 4.8% |
| Affine | 18.2 (31.8%) |
| HD | 34.8 (68.3%) |
| DBM | 38 (72%) |
| SS | 62.5% |
| NN | 78.3% |

## 3.2 One-shot generation of new examples

Not only can people classify new examples, they can generate new examples – even from just one image. While all generative classifiers can produce examples, it can be difficult to synthesize a range of compelling new examples in their raw form, especially since many models generate only features of raw stimuli (e.g, [5]). While DBMs [25] can generate realistic digits after training on thousands of examples, how well do these and other models perform from just a single training image?

We ran another Mechanical Turk task to produce nine new examples of 50 randomly selected handwritten character images from the evaluation set. Three of these images are shown in the leftmost column of Figure 6. After correctly answering comprehension questions, 18 participants in the USA were asked to "draw a new example" of 25 characters, resulting in nine examples per character. To simulate drawings from nine different people, each of the models generated nine samples after seeing exactly the same images people did, as described in Section SI-8 and shown in Figure 6. Low-level image differences were minimized by re-rendering stroke trajectories in the same way for the models and people. Since the HD model does not always produce well-articulated strokes, it was not quantitatively analyzed, although there are clear qualitative differences between these and the human produced images (Figure 6).

| Example | People | HBPL | Affine | HD |
|---|---|---|---|---|

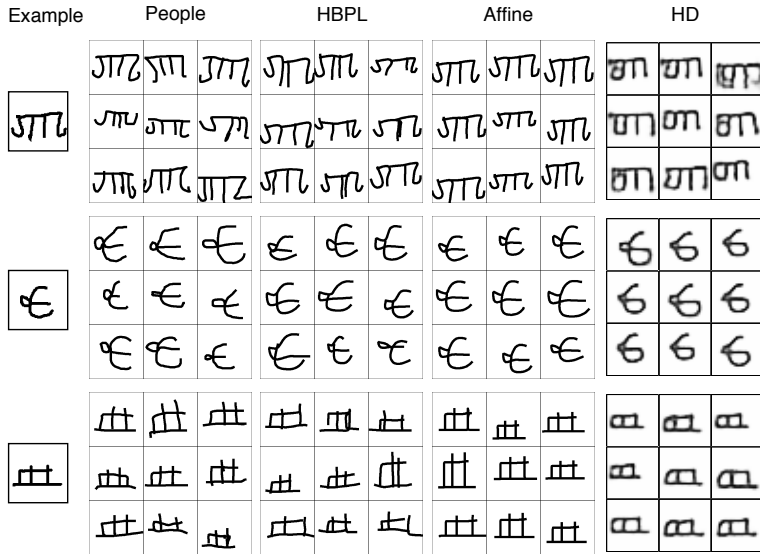

Figure 6: Generating new examples from just a single "target" image (left). Each grid shows nine new examples synthesized by people and the three computational models.

**Visual Turing test**. To compare the examples generated by people and the models, we ran a visual Turing test using 50 new participants in the USA on Mechanical Turk. Participants were told that they would see a target image and two grids of 9 images (Figure 6), where one grid was drawn by people with their computer mice and the other grid was drawn by a computer program that "simulates how people draw a new character." Which grid is which? There were two conditions, where the "computer program" was either HBPL or the Affine model. Participants were quizzed on their comprehension and then they saw 50 trials. Accuracy was revealed after each block of 10 trials. Also, a button to review the instructions was always accessible. Four participants who reported technical difficulties were not analyzed.

**Results**. Participants who tried to label drawings from people vs. HBPL were only 56% percent correct, while those who tried to label people vs. the Affine model were 92% percent correct. A 2-way Analysis of Variance showed a significant effect of condition ($p < .001$), but no significant effect of block and no interaction. While both group means were significantly better than chance, a subject analysis revealed only 2 of 21 participants were better than chance for people vs. HBPL, while 24 of 25 were significant for people vs. Affine. Likewise, 8 of 50 items were above chance for people vs. HBPL, while 48 of 50 items were above chance for people vs. Affine. Since participants could easily detect the overly consistent Affine model, it seems the difficulty participants had in detecting HBPL's exemplars was not due to task confusion. Interestingly, participants did not significantly improve over the trials, even after seeing hundreds of images from the model. Our results suggest that HBPL can generate compelling new examples that fool a majority of participants.

## 4 Discussion

Hierarchical Bayesian Program Learning (HBPL), by exploiting compositionality and causality, departs from standard models that need a lot more data to learn new concepts. From just one example, HBPL can both classify and generate compelling new examples, fooling judges in a "visual Turing test" that other approaches could not pass. Beyond the differences in model architecture, HBPL was also trained on the causal dynamics behind images, although just the images were available at evaluation time. If one were to incorporate this compositional and causal structure into a deep learning model, it could lead to better performance on our tasks. Thus, we do not see our model as the final word on how humans learn concepts, but rather, as a suggestion for the type of structure that best captures how people learn rich concepts from very sparse data. Future directions will extend this approach to other natural forms of generalization with characters, as well as speech, gesture, and other domains where compositionality and causality are central.

**Acknowledgments**

We would like to thank MIT CoCoSci for helpful feedback. This work was supported by ARO MURI contract W911NF-08-1-0242 and a NSF Graduate Research Fellowship held by the first author.

## Footnotes

[1] The number of spline evaluations is computed to be approximately 2 points for every 3 pixels of distance along the spline (with a minimum of 10 evaluations).

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
