[Supplementary Material]

# Supplementary material: One-shot learning by inverting a compositional causal process

## SI-1 Generating images of characters

Hierarchical Bayesian Program Learning (HBPL) is a generative model of characters, and this pseudocode goes through the steps of producing new character types and tokens. The first stochastic program GENERATETYPE samples a high level specification for a new character $\psi$ (Section 2.1), returning a handle to another stochastic program GENERATETOKEN. This second program can be run arbitrarily many times, each time producing a different (token) image $I^{(m)}$ of that character type (Section 2.2).

---

**procedure** GENERATETYPE
    $\kappa \leftarrow P(\kappa)$             ▷ Sample the number of strokes
    **for** $i = 1 \ldots \kappa$ **do**
        $z_i \leftarrow P(z_i)$          ▷ Sample the number and identities of the sub-strokes
        **for** $j = 1 \ldots n_i$ **do**
            $x_{ij} \leftarrow P(x_{ij}|z_{ij})$          ▷ Sample a sub-stroke's control points
            $y_{ij} \leftarrow P(y_{ij}|z_{ij})$          ▷ Sample a sub-stroke's scale
        **end for**
        $R_i \leftarrow P(R_i|z_1, ..., z_{i-1})$          ▷ Sample a stroke's relation to previous strokes
    **end for**
    $\psi \leftarrow \{\kappa, R, z, x, y\}$
    **return** @GENERATETOKEN$(\psi)$          ▷ Return the handle to a stochastic program
**end procedure**

---

**procedure** GENERATETOKEN$(\psi)$
    **for** $i = 1 \ldots \kappa$ **do**
        $R_i^{(m)} \leftarrow R_i$          ▷ Directly copy the type-level relation
        **if** $R_i^{(m)} =$ 'along' **then**
            $\tau_i^{(m)} \leftarrow P(\tau_i^{(m)}|\tau_i)$          ▷ Add variability to the attachment along the spline
        **end if**
        $L_i^{(m)} \leftarrow P(L_i^{(m)}|R_i^{(m)}, T_1^{(m)}, ..., T_{i-1}^{(m)})$          ▷ Sample stroke's starting location
        **for** $j = 1 \ldots n_i$ **do**
            $x_{ij}^{(m)} \leftarrow P(x_{ij}^{(m)}|x_{ij})$          ▷ Add variability to the control points
            $y_{ij}^{(m)} \leftarrow P(y_{ij}^{(m)}|y_{ij})$          ▷ Add variability to the sub-stroke scale
        **end for**
        $T_i^{(m)} \leftarrow f(L_i^{(m)}, x_i^{(m)}, y_i^{(m)})$          ▷ Compose a stroke's pen trajectory
    **end for**
    $A^{(m)} \leftarrow P(A^{(m)})$          ▷ Sample global image transformation
    $\epsilon^{(m)} \leftarrow P(\epsilon^{(m)})$          ▷ Sample the amount of pixel noise
    $\sigma_b^{(m)} \leftarrow P(\sigma_b^{(m)})$          ▷ Sample the amount blur
    $I^{(m)} \leftarrow P(I^{(m)}|T^{(m)}, A^{(m)}, \sigma_b^{(m)}, \epsilon^{(m)})$          ▷ Render and sample the binary image
    **return** $I^{(m)}$
**end procedure**

---

## SI-2 Probabilistic ink model

This section described how trajectories are rendered as an image (Section 2.2). After applying the transformation $A^{(m)}$ to the trajectories $T^{(m)}$, ink is placed along the trajectories using a grayscale ink model adapted from [1]. The continuous gray values of the pixels $0 \leq \rho_{ij} \leq 1$ are interpreted as probabilities of turning the pixels on. Each trajectory point contributes up to two "units" of ink to the four closest pixels using bilinear interpolation, where the ink units decrease linearly from 1 to 0 if two points are less than two pixel units apart. This method creates a thin line of ink, which is expanded out by convolving the image twice with the filter $b[a/12, a/6, a/12; a/6, 1 - a, a/6; a/12, a/6, a/12]$ and thresholding values greater than 1. The hyperparameters $a = 0.5$ and $b = 6$ were fit with maximum likelihood given a small subset of background image/parse pairs. The model also allows for variable levels of noise, $\sigma_b^{(m)} \sim \text{uniform}(0.5, 16)$ and $\epsilon^{(m)} \sim \text{uniform}(.0001, 0.5)$., which can ease search by encouraging partial solutions. The probability map is blurred by two convolutions with a Gaussian filter of size 11 with standard deviation $\sigma_b^{(m)}$. Finally, the probability of inking a binary pixel is a Bernoulli with probability $P(I_{ij} = 1) = (1 - \epsilon^{(m)})\rho_{ij} + \epsilon^{(m)}(1 - \rho_{ij})$, where $\epsilon^{(m)}$ can be interpreted as the probability of flipping a given pixel.

## SI-3 Identifying pauses in the drawing data

In the Omniglot dataset, the intervals of time at which the pen coordinates were sampled depend on a participant's web browser. Even within a single participant, the interval is irregular since only particular mouse events are tracked by the browser with a time stamp. Thus, all pen trajectories were normalized to have a fixed 50 millisecond sampling interval, which was approximated by linear interpolation. If the pen moved less than one pixel between two points, it was marked as a "pause." Sub-strokes are the segments extracted between pairs of pauses. For the purposes of learning the primitives, each sub-stroke trajectory was normalized to have zero mean and a range of 105 along its longest dimension. Sub-stroke trajectories with less than 5 time points were removed.

## SI-4 Learning high-level knowledge of motor programs

**Learning primitives.** A library of motor primitives, consisting of scale- and position-invariant movements terminated by a pause of the pen, was learned by clustering about 80,000 normalized sub-strokes in the background set (see Section SI-3 for identifying pauses in the drawings). Each sub-stroke was fit with a spline and re-represented by its control points in $\mathbb{R}^{10}$. A diagonal Gaussian Mixture Model (GMM) was used to partition sub-strokes into 1000 primitive elements, where the number of primitives was chosen via cross-validation (Figure 4a). Given this partition, the parameters for each primitive $z$, $\mu_z$, $\Sigma_z$, $\alpha_z$ and $\beta_z$, could be fit with maximum likelihood estimation (MLE). The transition probabilities between primitives $P(z_{ij}|z_{i(j-1)})$ were estimated by the smoothed empirical counts, where the regularization was chosen via cross-validation.

**Learning start positions and relations.** The distribution of stroke start positions $P(L_i)$ (Section 2.1) was estimated by discretizing the image plane and then fitting a separate grid model for a drawing's first, second, and third stroke (Figure 4c). All additional strokes share a single aggregated grid model. The probability of each cell was estimated from the empirical frequencies, and the complexity parameters for the grid granularity, smoothing, and aggregation threshold were chosen by cross-validation. Evidently, position is concentrated in the top-left (Figure 4c), where the concentration is stronger for earlier strokes. The other relational parameters, including mixing probabilities for the relation types $\theta_R$ and position noise $\Sigma_L$, were estimated by modeling start position as a mixture model over relations an then fitting the parameters with MLE.

**Learning token variability.** The token-level variability parameters $\{\sigma_x, \sigma_y, \sigma_\tau\}$ were less straightforward to estimate, since they cannot be directly computed from the motor data. Since these parameters control the variability of exemplars that are identical at the type-level, groups of highly-similar character exemplars were chosen based on shared primitives and stroke order. After global scale differences and outliers were removed, scale variability $\sigma_y$ was estimated from the deviations of each exemplar's scales from the mean values of the group. Attachment variability $\sigma_\tau$ was estimated

Figure SI-1: Illustration of extracting the character skeleton. a) Original image. b) Thinned image. c) Zoom highlights the imperfect detection of critical points (red pixels). d) Maximum circle criterion applied to the spurious critical points. e) Character graph after merging.

similarly. The same method estimates a value of shape variability $\sigma_x$ that is too large, so it was set by hand based on the visual appearance of the forward samples.

**Learning image parameters.** The distribution on global transformations $P(A^{(m)})$ was also learned from the background set. For each image, the center of mass and range of the inked pixels was computed. Second, images were grouped by character, and a transformation (scaling and translation) was computed for each image so that its mean and range matched the group average. Based on this large set of approximate transformations, a covariance on transformations $\Sigma_A$ could be estimated.

## SI-5    Inference

This section describes the inference algorithm (Section 2.4) in more detail. Probabilistic inference is very challenging in HBPL, since parsing an image requires searching a large combinatorial space of different strokes, relations, and sub-strokes. Fortunately, there has been decades of progress on developing bottom-up methods for analyzing the structure of handwritten characters. We take advantage of these algorithms, using a fast structural analysis to propose values of the latent variables in HBPL. This produces a large set of possible motor programs – each approximately fit to the image of interest. The most promising motor programs are chosen and refined with continuous optimization and MCMC. Each of these steps is explained in detail below.

### SI-5.1    Extracting the character skeleton

Search begins by applying a thinning algorithm to the raw image (Figure SI-1a) that reduces the line width to one pixel [2] (Figure SI-1b). This thinned image is used to produce candidate parses, although these parses are ultimately scored on the original image. The thinned image can provide an approximate structural analysis in the form of an undirected graph (as in Figure SI-1e), where edges (green) trace the ink and nodes (red) are placed at the terminal and fork (decision) points. While these decision points can be detected with simple algorithms [4], this process is imperfect and produces too many fork points (red pixels in Figure SI-1b and c). Many of these inaccuracies can be fixed by removing spurious branches and duplicate fork points with the "maximum circle criterion" [3]. This algorithm places the largest possible circle on each critical point, such that the circle resides within the original ink (gray regions in Figure SI-1d). All critical points with connecting circles are then merged (Figure SI-1e).

### SI-5.2    Generating random parses

A candidate parse is generated by a taking a random walk on the character skeleton with a "pen," visiting nodes until each edge has been traversed at least once. For many characters in the dataset, the graphs are sufficiently large that unbiased random walks do not explore the interesting parts of the parse space, which grows exponentially in the number of edges. Instead of an unbiased walk, the random walker stochastically prefers actions $A$ that minimize the local angle of the stroke trajectory around the decision point

$$P(A) \propto \exp(-\lambda\theta_A), \tag{SI-2}$$

Figure SI-2: Illustration of the random walk choosing between three potential moves, after drawing the topmost vertical edge (in the direction of the black arrow) and reaching a new decision point. The three potential trajectories are fit with the smoothest spline that stays within the image ink and does not deviate more than 3 pixels in any direction from the original trajectory (thick yellow line). Given these smoothed trajectory options, move a) has a local angle of 0 degrees (computed between the blue and purple vectors), move b) is 28 degrees, and move c) is 47 degrees.

where $\theta_A$ is the angle associated with action (Figure SI-2) and $\lambda$ is a constant. Two other possible actions, picking up the pen and re-tracing a trajectory, pay a cost of 45 and 90 degrees respectively. If the pen is in lifted position, the random walk must pick a node to put the pen down on to start the next stroke. To bias the random walk towards completing the drawing efficiently, the start node is chosen in proportion to $1/b^\gamma$, where $b$ is the number of new (unvisited) edges branching from that node.

This random walk process is repeated many times to generate a range of candidate parses. Random walks are generated until 150 parses or 100 unique strokes, shared across all of the parses, have been sampled. Limiting the number of unique strokes is a natural criterion, since sub-parsing these strokes is a computational bottleneck, as described in the next section. Larger values of the constants $\lambda$ and $\gamma$ are better for parsing complex characters, since low stochasticity is critical for finding smooth parses in a tremendous search space. But smaller values of $\lambda$ and $\gamma$ are better for simple characters, where the algorithm has the computational resources to more exhaustively explore the parse space. To get the best of both, different values of $\lambda$ and $\gamma$ are sampled before starting each random walk, producing both low and high entropy random walks as candidates.

### SI-5.3 Searching for sub-strokes

Before any candidate parse can be scored as a complete motor programs (Eq. 5), the strokes must be sub-divided into sub-strokes. To do so, the strokes in each random walk are smoothed while enforcing that the trajectories stay within the original ink (as in Figure SI-2), in order to correct for spurious curves that arise from thinning algorithms (see Figure SI-2a for an example). The smoothed strokes are then parsed into sub-strokes by running a simple greedy search for each stroke trajectory. During search, operators add, remove, perturb, or replace pauses along the trajectory to form sub-strokes. To score the quality of the decomposition, the sub-strokes are fit with splines, classified as primitives $z_i$, and scored by the generative model for strokes

$$ P(x_i^{(m)}, y_i^{(m)}, z_i) = P(z_i) \prod_{j=1}^{n_i} P(y_{ij}^{(m)}|y_{ij}) P(y_{ij}|z_{ij}) \int P(x_{ij}^{(m)}|x_{ij}) P(x_{ij}|z_{ij}) \, \mathrm{d}x_{ij}, \quad \text{(SI-2)} $$

where $y_i$ is approximated by setting it equal to $y_i^{(m)}$. There is also a hard constraint that the spline approximation to the original trajectory can miss its target by no more than 3 pixels.

After the search process is run for each stroke trajectory, each candidate motor program with variables $\psi$ and $\theta^{(m)}$ is fully-specified and tracks the image structure relatively closely. Thus, the prior score $P(\theta^{(m)}|\psi)P(\psi)$ is used to select the $K$ best candidates to progress to the next stage of search, which fine-tunes the motor programs.

## SI-5.4 Optimization and fine-tuning

Holding the discrete variables fixed, the set of continuous variables (including $L^{(m)}, \tau^{(m)}, x^{(m)}, y^{(m)}, \epsilon^{(m)}, \sigma_b^{(m)}$) are optimized to fit the pixel image with Matlab's "active-set" constrained optimization algorithm, using the full generative score as the objective function (Eq. 5). There are two simplifications to reduce the number of variables: the affine warp $A^{(m)}$ is disabled and the relations $R_i$ are left unspecified and re-optimized during each evaluation of the objective function. After optimization finds a local maximum, the optimal joint setting of stroke directions and stroke order are chosen using exhaustive enumeration for characters with five strokes or less, while considering random subsets for more complex characters. Finally, the best scoring relations are chosen, and a greedy search to split strokes (at any sub-stroke transition) and merge strokes (at places where a stroke begins at the end of the previous stroke) proceeds until the score can no longer be improved.

## SI-5.5 MCMC to estimate local variance

At this step, the algorithm has $K$ high-probability parses $\psi^{[1]}, \theta^{(m)[1]}, ..., \psi^{[K]}, \theta^{(m)[K]}$ which have been fine-tuned to the images. Each parse spawns a separate run of MCMC to estimate the local variance around the type-level by sampling from $P(\psi|\theta^{(m)[i]})$. This is inexpensive since it does not require evaluating the likelihood of the image. Metropolis Hastings moves with simple Gaussian proposals are used for the shapes $x$, scales $y$, global positions $L$, and attachments $\tau$. The sub-stroke ids $z$ are updated with Gibbs sampling. Each chain is run for 200 iterations over variables, and $N = 10$ linearly spaced samples across the chain are stored to form the $Q(\cdot)$ approximation to the posterior in Eq. 6.

# SI-6 The characters in each one-shot classification task

One-shot classification was tested on 10 within-alphabet classification tasks, where the alphabets were chosen from the evaluation set (Section 3.1). Each task's images were produced by four relatively typical drawers, and the set of 20 characters was picked to maximize diversity when alphabets had more than 20 characters. The four drawers were randomly paired to form two groups, and one drawer in each group provided the test examples for 20 trials while the other drawer provided the 20 training examples for each of these trials.

# SI-7 One-shot classification

One-shot classification involves computing the posterior predictive distribution of $P(I^{(T)}|I^{(c)})$ for a test image $I^{(T)}$ given a training image $I^{(c)}$. This section shows the derivation of the approximation that we use to compute the score (see Eq. 8 and Eq. 9 in Section 3.1)

$$
\begin{aligned}
P(I^{(T)}|I^{(c)}) &= \int P(I^{(T)}, \theta^{(T)}, \theta^{(c)}, \psi|I^{(c)}) \, \mathrm{d}(\psi, \theta^{(c)}, \theta^{(T)}) \\
&= \int P(I^{(T)}|\theta^{(T)}) P(\theta^{(T)}, \theta^{(c)}, \psi|I^{(c)}) \, \mathrm{d}(\psi, \theta^{(c)}, \theta^{(T)}) \\
&= \int P(I^{(T)}|\theta^{(T)}) [\int P(\theta^{(T)}|\psi) P(\theta^{(c)}, \psi|I^{(c)}) \, \mathrm{d}(\psi, \theta^{(c)})] \, \mathrm{d}\theta^{(T)} \\
&\approx \int P(I^{(T)}|\theta^{(T)}) [\int P(\theta^{(T)}|\psi) Q(\theta^{(c)}, \psi, I^{(c)}) \, \mathrm{d}(\psi, \theta^{(c)})] \, \mathrm{d}\theta^{(T)} \\
&= \int P(I^{(T)}|\theta^{(T)}) [\sum_{i=1}^{K} \frac{w_i}{N} \sum_{j=1}^{N} P(\theta^{(T)}|\psi^{[ij]})] \, \mathrm{d}\theta^{(T)} \\
&= \sum_{i=1}^{K} w_i \int P(I^{(T)}|\theta^{(T)}) \frac{1}{N} \sum_{j=1}^{N} P(\theta^{(T)}|\psi^{[ij]}) \, \mathrm{d}\theta^{(T)} \\
&\approx \sum_{i=1}^{K} w_i \max_{\theta^{(T)}} P(I^{(T)}|\theta^{(T)}) \frac{1}{N} \sum_{j=1}^{N} P(\theta^{(T)}|\psi^{[ij]}).
\end{aligned}
$$

## SI-8   One-shot generation of new examples

This section describes how each model generates new examples, given a single reference image.

**HBPL.** HBPL is tasked with generating a new example image $I^{(2)}$ given another image $I^{(1)}$, and thus, it is desirable to produce samples from $P(I^{(2)}, \theta^{(2)}|I^{(1)})$. Utilizing the same discrete approximation $Q(\cdot)$ as with classification (Eq. 6), we derive a distribution that is straightforward to sample from:

$$
\begin{aligned}
P(I^{(2)}, \theta^{(2)}|I^{(1)}) =\ & \int P(I^{(2)}, \theta^{(2)}|\theta^{(1)}, \psi) P(\theta^{(1)}, \psi|I^{(1)})\, \mathrm{d}(\psi, \theta^{(1)}) \\
=\ & \int P(I^{(2)}|\theta^{(2)}) P(\theta^{(2)}|\psi) P(\theta^{(1)}, \psi|I^{(1)})\, \mathrm{d}(\psi, \theta^{(1)}) \\
\approx\ & \int P(I^{(2)}|\theta^{(2)}) P(\theta^{(2)}|\psi) Q(\theta^{(1)}, \psi, I^{(1)})\, \mathrm{d}(\psi, \theta^{(1)}) \\
=\ & \sum_{i=1}^{K} \sum_{j=1}^{N} \frac{w_i}{N} P(I^{(2)}|\theta^{(2)}) P(\theta^{(2)}|\psi^{[ij]}).
\end{aligned}
$$

The HBPL inference algorithm was run to collect $K = 10$ parses of the image $I^{(1)}$. When using the above formulation directly, the model would repeatedly sample just the best-scoring parse in most cases, since even small differences in the parses can lead to massive differences in weights $w_i$ due to the high-dimensional raw data. To avoid dramatically underestimating the variety of parses in the posterior, the weights $w_i$ were set to be inversely proportional to their rank order $1/\sigma(i)$, where $\sigma(\cdot)$ is the permutation function, or rank of the $i$th parse when sorted from highest to lowest score. The new sampling distribution is then

$$
I^{(2)}, \theta^{(2)}|I^{(1)} \sim \frac{1}{\sum_{i=1}^{K} i^{-1}} \sum_{i=1}^{K} \frac{1}{\sigma(i)} \sum_{j=1}^{N} \frac{1}{N} P(I^{(2)}|\theta^{(2)}) P(\theta^{(2)}|\psi^{[ij]}).
$$

To minimize superficial differences between the model's samples and people's drawings, in terms of scale or low-level image differences, stroke trajectories in both cases were plotted using the same deterministic renderer. Also, rather than sampling an image transformation $A^{(2)}$, it was manually set to match a human drawing of the same character.

**Affine model.** All of the above steps for HBPL were followed, except that none of the token-level variables are resampled, meaning that $P(\theta^{(2)}|\psi^{[ij]}) = \delta(\theta^{(2)} - \theta^{(1)[i]})$.

**HD.** Given a single example of a new character, the model quickly approximately infers which super-class the new character belongs to. Given the super-class parameters, the model samples the states of the top-level DBM's features from the HDP prior, followed by computing grayscale pixel values for the bottom layer of the DBM.