[Reviews · NeurIPS 2013]

Submitted by Assigned_Reviewer_4

The manuscript reports a system that learns about an image of a 2-D shape from a single exposure. The system is pre-trained with a large number of other images. I like this paper because it combines a large number of unusual (though not necessarily novel) ideas and approaches. For instance, I like: studying the problem of one-shot learning, the set of images used in this paper, the identification and use of "motor primitives" to represent sub-strokes, the use of spatial relations among sub-strokes to represent larger strokes, the way that parsing was conducted, etc. In isolation, any one of these aspects that I like might not be that impressive. But considered in combination, the overall research project is very appealing.

My best suggestion for improving this manuscript is that the authors do a better job of explicitly identifying the key contributions that the reader should learn from this study. Is the power of the system mostly due to its use of motor primitives? Or is the power mostly due to the combinatorial nature of the primitives? Or is the power mostly due to the parsing algorithm? Or is the power mostly due to Bayesian inference? When building future systems, what elements of the current system are crucial and thus should be incorporated in future designs?
Summary: In summary, a nice package of ideas, elegantly implemented.

Submitted by Assigned_Reviewer_5

Reviewer response to rebuttal:

I have read through the author's rebuttal and I am happy with the proposed changes. I have not changed my review as I already recommended this paper for acceptance.

Previous Review:

In this work, the authors develop a hierarchical generative model for producing and classifying written characters with the goal of achieving a high level of performance with just one training example. The model is rooted in learning the compositional structure of characters and the causal relationship that dictates how characters are produced. The model is compared to a simpler version of the model that does not represent character strokes, a deep boltzmann machine approach, and a hierarchical deep learning method. In addition to significantly outperforming these comparison models, the model achieves an impressive error rate that is very similar to human classification error rate. Furthermore, the authors demonstrate that the model can generate realistic character drawings after receiving only one training example as tested by a “Visual Turing Test” where participants were unable to reliably distinguish human drawn characters from machine drawn characters. The success of this approach appears to be attributed to its ability to learn from the richer compositional and causal properties of one image.


COMMENTS TO AUTHORS:

The near-human level of performance in a one-shot learning task in this work is very impressive and suggests that there is potentially great value in exploiting compositionality and causality in this domain. My only complaint is that the model is very complex which makes it a bit difficult to identify where the big win is. However, the comparison to the similar Affine model helps with this problem.

It would be nice to see this model compared directly to results presented in previous work (maybe [25]).

MINOR COMMENTS:

I had trouble finding reference [25] (R. Salakhutdinov, J. B. Tenenbaum, and A. Torralba. Hierarchical deep models for one-shot learning. In Neural Information Processing Systems (NIPS), 2011.)
Perhaps one of these was the intended reference?

R. Salakhutdinov, J. Tenenbaum , A. Torralba. Learning to Learn with Compound Hierarchical-Deep Models. NIPS, 2011.

OR

R. Salakhutdinov, J. Tenenbaum, and A. Torralba . One-Shot Learning with a Hierarchical Nonparametric Bayesian Model. JMLR WC&P Unsupervised and Transfer Learning, 2012.
Summary: Solid paper with impressive results. Complexity of the model is a bit hard to follow but everything seems sound.

Submitted by Assigned_Reviewer_7

This paper presents a system for one shot learning of visual concepts based on "hierarchical Bayesian program learning". The system outperforms several standard machine learning methods on the visual concept learning task defined by the authors.

This is high-quality work, generally clearly presented, and likely to be of interest to the NIPS audience. However, there are some issues with the paper. The main one is that it is extremely similar to previous work (references 15 and 16 in the paper), and it is hard to assess whether the approach is a significant improvement over this previous work since the methods presented in the previous work are never used for comparison. Minimally, the paper needs to acknowledge the previous work, explain how the present approach differs, and compare to the previous results. Since the previous results also showed improvements over standard machine learning methods on a similar but not identical task, it is hard to tell whether the performance shown here is better.

The designation of the approach as "hierarchical Bayesian program learning" might be a little misleading, or oversell the approach, as it's not really doing program induction. It is performing probabilistic inference in a structured probabilistic model, where the elements of the model correspond to the steps in a motor program. The presentation makes it seem like the approach is more general than this.

The task used to motivate and evaluate the approach is one that has only been explored in the previous papers mentioned. It would be valuable to show that this approach also results in improvements in performance on other more standard machine learning task. A skeptic could say that the results aren't particularly impressive, as they demonstrate that carefully engineering a system to solve a specific problem results in an improvement over generic machine learning methods, which isn't really news. In order for the approach to have a high impact it needs to be shown to be applicable beyond this narrow domain.

A minor point, but the Affine model should be explained more clearly.
Summary: Overall, I think this is a strong paper with impressive results. However, it is not clear how much these results improve on previous work as the relationship to previous work is not discussed, and the impact could be increased by covering a broader range of problems.
Author Feedback

Author rebuttal: We would like to thank all reviewers for their thoughtful feedback on our work. We will incorporate these suggestions in the final version.

We agree with Reviewer 3 that the paper should directly compare the Hierarchical Bayesian Program Learning (HBPL) model with the related model published in references [15] and [16]. This was an oversight, and a direct comparison will be added to the final version. The current work is a significant advance over the previous work, not only in classification performance, but also in capturing the richness of the causal process behind handwritten characters. If the previous model [15,16] was asked to generate multiple examples of a single character, the examples would look nearly identical with just slight differences in the position of the (otherwise rigid) strokes. Furthermore, the model in [15,16] does not represent the order of the strokes or their relations. Because of these differences, the previous model achieved 45.1% error rate on 20-way one-shot classification while HBPL achieved a 4.8% error rate on a more difficult within-alphabet 20-way classification task. To make the comparison more direct, we will get results for the model in [15,16] on this more difficult task for the final version, although we expect performance to be no better than 45% errors.

On a related note, Reviewer 2 wrote "It would be nice to see this model compared directly to results presented in previous work (maybe [25])," where [25] is the reference that introduced the Hierarchical Deep (HD) model for one-shot learning. We want to thank this reviewer for pointing out the mistake in the reference's title, and we also want to clarify that we did compare directly with the HD model on our one-shot learning task.

It seems that Reviewers 1 and 2 would be curious to see a more detailed analysis of the model components and how they affect performance. This is valuable feedback. Much of this will require a longer journal paper, since we think that multiple components contribute importantly to results, based on our experience in building the model. We want to show this using various manipulations and lesioning of the background training data and model parameters.

Reviewer 3 also pointed out that the deep learning models we directly compared with [Hierarchical Deep (HD) and Deep Boltzmann Machines (DBM)] are more general than HBPL. He/she also suggested exploring tasks from other domains with HBPL. This is an important suggestion, although we want to emphasize that our focus is slightly different than standard machine learning; we are trying to capture the knowledge that people bring to bear when learning a new concept. This requires a more diverse set of evaluations than is typical in machine learning, as well as more domain specific knowledge. Nonetheless, we hope that the approach will prove useful to a machine learning audience beyond this specific domain, and we agree with Reviewer 3 that the work could be applied to other types of concepts. We are currently working on applications in the domains of speech, gesture, and other visual concepts.